World of Crayfish™: a web platform towards real-time global mapping of freshwater crayfish and their pathogens

Ion Mihaela C. 1
Bloomer Caitlin C. 2
Bărăscu Tudor I. 3
Oficialdegui Francisco J. 4
Shoobs Nathaniel F. 5
Williams Bronwyn W. 6
Scheers Kevin 7
Clavero Miguel 8
Grandjean Frédéric 9
Collas Marc 10
Baudry Thomas 9
Loughman Zachary 11
Wright Jeremy J. 12
Ruokonen Timo J. 13
Chucholl Christoph 14
Guareschi Simone 15
Koese Bram 16
Banyai Zsombor M. 17 18
Hodson James 19
Hurt Margo 20
Kaldre Katrin 20
Lipták Boris 4 21
Fetzner James W. 22
Cancellario Tommaso 23
Weiperth András 24
Birzaks Jạnis 25
Trichkova Teodora 26
Todorov Milcho 26
Balalaikins Maksims 25
Griffin Bogna 27
Petko Olga N. 28
Acevedo-Alonso Ada 29
D’Elía Guillermo 30
Śliwińska Karolina 31
Alekhnovich Anatoly 31
Choong Henry 32
South Josie 33 34
Whiterod Nick 35 36
Zorić Katarina 37
Haase Peter 38 39
Soto Ismael 4
Brady Daniel J. 40
Haubrock Phillip J. 4 39 41
Torres Pedro J. 42
Şadrin Denis 28
Vlach Pavel 43
Kaya Cüneyt 44
Woo Jung Sang 45
Kim Jin-Young 46
Vermeersch Xavier H.C. 47
Bonk Maciej 48
Guiaşu Radu 49
Harlioğlu Muzaffer M. 50
Devlin Jane 51
Kurtul Irmak 52 53
Błońska Dagmara 54
Boets Pieter 55
Masigol Hossein 28
Cabe Paul R. 56
Jussila Japo 57
Vrålstad Trude 58
Beresford David V. 59
Reid Scott M. 51
Patoka Jiří 60
Strand David A. 58
Tarkan Ali S. 54 61
Steen Frédérique 7
Abeel Thomas 62
Harwood Matthew 34
Auer Samuel 63
Kelly Sandor 64
Giantsis Ioannis A. 65 66
Maciaszek Rafał 67
Alvanou Maria V. 65
Aksu Önder 68
Hayes David M. 69
Kawai Tadashi 70
Tricarico Elena 71
Chakandinakira Adroit 72
Barnett Zanethia C. 73
Kudor Ştefan G. 74
Beda Andreea E. 75
Vîlcea Lucian 76
Mizeranschi Alexandru E. 28 77
Neagul Marian 28
Licz Anton 78
Cotoarbă Andra D. 79
Petrusek Adam 80
Kouba Antonín 4
Taylor Christopher A. 2
Pârvulescu Lucian lucian.parvulescu@e-uvt.ro 28 79
1 Institute of Biology Bucharest, Romanian Academy , Bucharest , Romania
2 Illinois Natural History Survey, Prairie Research Institute , Champaign , IL , United States of America
3 Qtibia Engineering , Râmnicu-Vâlcea , Romania
4 Faculty of Fisheries and Protection of Waters, South Bohemian Research Centre of Aquaculture and Biodiversity of Hydrocenoses, University of South Bohemia in České Budějovice , Vodňany , Czech Republic
5 Museum of Biological Diversity, Department of Evolution, Ecology, and Organismal Biology, The Ohio State University , Columbus , OH , United States of America
6 Research Laboratory, North Carolina Museum of Natural Sciences , Raleigh , NC , United States of America
7 Unit Freshwater habitats, Research Institute for Nature and Forest , Brussels , Belgium
8 Estación Biológica de Doñana–CSIC , Sevilla , Spain
9 Laboratoire Ecologie et Biologie des Interactions - UMR CNRS 7267, Laboratoire EBI - Equipe Ecologie Evolution Symbiose - Batiment B31, Université de Poitiers , Poitiers , France
10 Office Français de la Biodiversité , Epinal , France
11 Department of Organismal Biology, Ecology, and Zoo Science, West Liberty University , West Liberty , WV , United States of America
12 New York State Museum , Albany , NY , United States of America
13 Natural Resources Institute Finland , Jyväskylä , Finland
14 Fisheries Research Station Baden-Württemberg , Langenargen , Germany
15 Department of Life Sciences and Systems Biology, University of Turin , Turin , Italy
16 Naturalis Biodiversity Center , Leiden , Netherlands
17 Doctoral School of Environmental Science, Hungarian University of Agriculture and Life Sciences , Gödöllő , Hungary
18 Department of Freshwater Fish Ecology, Institute of Aquaculture and Environmental Safety, Hungarian University of Agriculture and Life Sciences , Gödöllő , Hungary
19 School of Biology, Faculty of Biological Science, University of Leeds , Leeds , United Kingdom
20 Chair of Aquaculture, Institute of Veterinary Medicine and Animal Sciences, Estonian University of Life Sciences , Tartu , Estonia
21 Slovak Environment Agency , Banská Bystrica , Slovakia
22 Section of Invertebrate Zoology, Carnegie Museum of Natural History , Pittsburgh , PA , United States of America
23 Balearic Biodiversity Centre, Department of Biology, University of the Balearic Islands , Palma , Spain
24 Department of Systematic Zoology and Ecology, Eötvös Loránd University , Budapest , Hungary
25 Institute of Life Sciences and Technology, Department of Biodiversity, Daugavpils University , Daugavpils , Latvia
26 Institute of Biodiversity and Ecosystem Research, Bulgarian Academy of Sciences , Sofia , Bulgaria
27 Fish Health Unit, Marine Institute , Galway , Ireland
28 Crayfish Research Centre, Institute for Advanced Environmental Research, West University of Timisoara , Timisoara , Romania
29 Independent Researcher , Tunja-Boyacá , Colombia
30 Instituto de Ciencias Ambientales y Evolutivas, Universidad Austral de Chile , Valdivia , Chile
31 Scientific and Practical Center for Biological Resources of the National Academy of Science of Belarus , Minsk , Belarus
32 Royal British Columbia Museum , Victoria , British Columbia , Canada
33 South African Institute for Aquatic Biodiversity , Makhanda , South Africa
34 Water@Leeds, School of Biology, Faculty of Biological Sciences , Leeds , United Kingdom
35 Nature Glenelg Trust , South Australia , Australia
36 CLLMM Research Centre, Goyder Institute for Water Research , Goolwa , South Australia , Australia
37 Department of Hydroecology and Water Protection, Institute for Biological Research “Siniša Stanković”, National Institute of the Republic of Serbia, University of Belgrade , Belgrade , Serbia
38 Faculty of Biology, University of Duisburg-Essen , Essen , Germany
39 Department of River Ecology and Conservation, Senckenberg Research Institute and Natural History Museum Frankfurt , Gelnhausen , Germany
40 Branch for Bioresources, Fraunhofer Institute for Molecular Biology and Applied Ecology , Gießen , Germany
41 CAMB, Center for Applied Mathematics and Bioinformatics, Gulf University for Science and Technology , Hawally , Kuwait
42 Biology Department, College of the Holy Cross , Worcester , MA , United States of America
43 Center of Biology, Geosciences and Environmental Education, Faculty of Education, University of West Bohemia , Plzeň , Czech Republic
44 Faculty of Fisheries, Recep Tayyip Erdogan University , Rize , Turkey
45 DASARI Research Institute of BioResources , Daejeon , Republic of Korea
46 Research Center for Endangered Species, National Institute of Ecology , Yeongyang , Republic of Korea
47 Department Nature and Biodiversity, Brussels Environment , Brussels , Belgium
48 Institute of Nature Conservation, Polish Academy of Sciences , Kraków , Poland
49 Biology Program, Glendon College, York University , Toronto , Ontario , Canada
50 Fisheries Faculty, Fırat University , Elazığ , Turkey
51 Ontario Ministry of Natural Resources , Peterborough , Ontario , Canada
52 Marine and Inland Waters Sciences and Technology Department, Faculty of Fisheries, Ege University , Izmir , Turkey
53 Department of Life and Environmental Sciences, Faculty of Science and Technology, Bournemouth University , Poole, Bournemouth , United Kingdom
54 Department of Ecology and Vertebrate Zoology, Faculty of Biology and Environmental Protection, University of Lodz , Łódź , Poland
55 Provincial Centre of Environmental Research , Ghent , Belgium
56 Biology Department, Washington and Lee University , Lexington , VA , United States of America
57 Department of Environmental and Biological Sciences, University of Eastern Finland , Kuopio , Finland
58 Department of Aquatic Animal Health, Norwegian Veterinary Institute , Ås , Norway
59 Biology and Trent School of the Environment, Trent University , Peterborough , Ontario , Canada
60 Department of Zoology and Fisheries, Faculty of Agrobiology, Food and Natural Resources, Czech University of Life Sciences Prague , Prague , Czech Republic
61 Department of Basic Sciences, Faculty of Fisheries, Muğla Sıtkı Koçman University , Muğla , Turkey
62 Agro- and Biotechnology, Odisee University of Applied Sciences , Sint-Niklaas , Belgium
63 Blattfisch e.U. , Wels , Austria
64 University of Central Florida , Orlando , FL , United States of America
65 Department of Animal Science, Faculty of Agricultural Sciences, University of Western Macedonia , Florina , Greece
66 Laboratory of Ichthyology & Fisheries, Faculty of Agriculture, Forestry and Natural Environment, Aristotle University of Thessaloniki , Thessaloniki , Greece
67 Department of Animal Genetics and Conservation, Institute of Animal Sciences, Warsaw University of Life Sciences , Warsaw , Poland
68 Department of Aquaculture of Fisheries Faculty, Munzur University , Tunceli , Turkey
69 Eastern Kentucky University , Richmond , KY , United States of America
70 Hokkaido Research Organization, Central Fisheries Research Institute , Yoichi Hokkaido , Japan
71 Dipartimento di Biologia, Università di Firenze , Sesto Fiorentino , Italy
72 Lake Kariba Fisheries Research Institute, Zimbabwe Parks and Wildlife Management Authority , Kariba , Zimbabwe
73 USDA Forest Service, Southern Research Station, Center for Bottomland Hardwoods Research , Clemson , SC , United States of America
74 “Simion Mehedinţi - Nature and Sustainable Development” Doctoral School, University of Bucharest , Bucharest , Romania
75 Department of Computer Science and Information Technology, Faculty of Engineering in Foreign Languages, University Politehnica of Bucharest , Bucharest , Romania
76 Department of Economic Informatics and Cybernetics, Bucharest University of Economic Studies , Bucharest , Romania
77 Research and Development Station for Bovine - Arad , Arad , Romania
78 Information Technology & Communications Department, West University of Timisoara , Timisoara , Romania
79 Department of Biology, Faculty of Chemistry, Biology, Geography, West University of Timisoara , Timisoara , Romania
80 Department of Ecology, Faculty of Science, Charles University , Prague , Czech Republic
Korla Praveen
Electronic publication date: 2024 Oct 14
Publication date: 2024
Volume: 12
Electronic Location ID: e18229
Received 2024 May 1; Accepted 2024 Sep 13
Copyright: ©2024 Ion et al.
Copyright year: 2024
Copyright holder: Ion et al.
License: This is an open access article distributed under the terms of the Creative Commons Attribution License, which permits unrestricted use, distribution, reproduction and adaptation in any medium and for any purpose provided that it is properly attributed. For attribution, the original author(s), title, publication source (PeerJ) and either DOI or URL of the article must be cited.
License URL: https://creativecommons.org/licenses/by/4.0/

Keywords: Aphanomyces astaci, Endangered species, Invasive species, Open data, Astacidae, Cambaridae, Species distribution, Parastacidae

Funding: West University of Timisoara through the U InnoVaTe programme Ministry of Research, Innovation and Digitization, CNCS/CCCDI–UEFISCDI, within the PNCDI III programme No. PN-III-P4-ID-PCE-2020-1187 This work was supported by (1) the West University of Timisoara through the U InnoVaTe programme, via an UIVT Mature grant awarded to the, World of Crayfish (WoC1.0)” project in May 2023, and (2) the Ministry of Research, Innovation and Digitization, CNCS/CCCDI–UEFISCDI, within the PNCDI III programme (No. PN-III-P4-ID-PCE-2020-1187). The funders had no role in study design, data collection and analysis, decision to publish, or preparation of the manuscript.

==============================
Freshwater crayfish are amongst the largest macroinvertebrates and play a keystone role in the ecosystems they occupy. Understanding the global distribution of these animals is often hindered due to a paucity of distributional data. Additionally, non-native crayfish introductions are becoming more frequent, which can cause severe environmental and economic impacts. Management decisions related to crayfish and their habitats require accurate, up-to-date distribution data and mapping tools. Such data are currently patchily distributed with limited accessibility and are rarely up-to-date. To address these challenges, we developed a versatile e-portal to host distributional data of freshwater crayfish and their pathogens (using Aphanomyces astaci, the causative agent of the crayfish plague, as the most prominent example). Populated with expert data and operating in near real-time, World of Crayfish™ is a living, publicly available database providing worldwide distributional data sourced by experts in the field. The database offers open access to the data through specialized standard geospatial services (Web Map Service, Web Feature Service) enabling users to view, embed, and download customizable outputs for various applications. The platform is designed to support technical enhancements in the future, with the potential to eventually incorporate various additional features. This tool serves as a step forward towards a modern era of conservation planning and management of freshwater biodiversity.

Introduction

Accurate mapping of species distributions is essential for enhancing and improving conservation efforts and decision-making regarding both native and non-native taxa (Gonzalez et al., 2023). Data derived models, such as species distribution models, are generally helpful for predicting various scenarios of current and future distributions but require specific expertise and high-quality data for model training and calibration (Liu et al., 2020; Guareschi et al., 2024). Large, globally open access datasets are a priceless resource for tracking changes in the world’s biodiversity (Liang & Gamarra, 2020). Easy-to use and trustworthy platforms increase the capacity of prompt data collection and adequate assessments of growing pressures on many species in a variety of ecosystems (Pyšek et al., 2020). Meanwhile, the computation of global data descriptors for geo-physical, climate, and environmental parameters with high-resolution data has increased in availability, with software even designed to provide time series spanning from the past to the present day (Soto et al., 2023). When high quality environmental and biological data are combined (Jetz et al., 2019), the potential for highly powerful modeling approaches can be realized (Domisch, Amatulli & Jetz, 2015; Beck et al., 2018; Kaufman et al., 2020) including, for example, spatially explicit conservation assessment and planning (Pârvulescu et al., 2020; Ion et al., 2024), predicting future trajectories of biological invasions and monitoring disease outbreaks across river networks (Satmari et al., 2023; Soto et al., 2023).

The reliability and robustness of modeling approaches highly depend on accurate and precise species distribution data (Rocchini et al., 2011; Brooks et al., 2019; Li, Bearup & Liao, 2020). Data found in publicly available databases have to be treated with care due to different biases and sources of uncertainties, e.g., the increasing data collection by citizen scientists with varying levels of expertise or variable data-gathering methodologies that challenge the reliability and comparability of the datasets (Kosmala et al., 2016; Callaghan et al., 2020). Moreover, inaccuracies in species identification and potential disturbances to legally protected or vulnerable species by untrained individuals can compromise data quality and further contribute to conservation concerns, such as the proliferation of pathogens through improper biosecurity during monitoring (Barve et al., 2011; Robinson et al., 2020; Lipták et al., 2024).

Freshwater crayfish (Crustacea: Decapoda: Astacida) (sensu Scholtz & Richter, 1995; Crandall & De Grave, 2017) naturally occur on all continents except Antarctica and continental Africa and include roughly 700 described species (Crandall & Buhay, 2007; Crandall & De Grave, 2017) with distinct diversity hotspots in southeastern North America and Australasia (Lodge et al., 2012). In freshwater ecosystems, crayfish play an integral role due to their size, longevity, abundance, and omnivorous feeding habits (Momot, 1995). Furthermore, crayfish are regarded as a classical conservation conundrum where many are threatened and vulnerable in their native ranges, whereas others are recognized as damaging biological invaders (Momot, 1995). Understanding the drivers of this conundrum requires accurate information on their global distributions accompanied by environmental mapping, especially for non-native crayfish, which, once introduced, can become a major pressure on freshwater biota and the functioning of aquatic ecosystems (Emery-Butcher, Beatty & Robson, 2020). Crayfish invasions pose a significant threat to native crayfish species due to competition, predation, and the transmission of pathogens. The most prominent pathogen, with severe impacts on Eurasian crayfish populations, is Aphanomyces astaci Schikora, the agent responsible for crayfish plague outbreaks in crayfish of non-North American origin (Chucholl & Schrimpf, 2016; Jussila & Edsman, 2020). Other crayfish pathogens (e.g., Grandjean et al., 2019; Stratton et al., 2023; Zingre et al., 2023) remain much less studied, as their impacts and dispersal have not been notably detrimental to crayfish (Longshaw, 2011). As the eradication of non-native crayfish is often deemed impossible (Lidova et al., 2019; Manfrin et al., 2019), the prevention of introductions and early detection of introduced non-native crayfish remain the most effective methods to avoiding and mitigating invasions (Reaser et al., 2019).

Tracking native and non-native crayfish distributions is therefore of the utmost importance for the conservation of both crayfish and the variety of aquatic ecosystems they inhabit (Jussila et al., 2021). Without rapid and accessible communication of field observations, along with geo-spatial descriptive information, effective early management actions are impossible (Sax & Gaines, 2003; Mcclenachan, Ferretti & Baum, 2012; Salomaki et al., 2020; Miller et al., 2021). Accurate species identification for many crayfish species, however, remains challenging, particularly in sympatric areas, with expert studies and reports remaining the most trustworthy sources for occurrence data (Costello, 2009; Costello et al., 2013; Costello & Wieczorek, 2014). The value of such reliable expert sources may not be maximized unless they are digitized and made available in centralized databases (Boakes et al., 2010). Indeed, although the distributions of both native and non-native crayfish as well as the crayfish plague pathogen can be scattered in peer-reviewed publications and other verified resources, there is still a lack of combined, reliable, and publicly accessible data, ultimately limiting the practical applications for researchers and conservation managers. Traditional mapping has been useful for documenting the locations of crayfish species, both native and non-native, and the distribution of different genotypes of the crayfish plague pathogen across Europe (Kouba, Petrusek & Kozák, 2014; Ungureanu et al., 2020). However, these maps quickly become outdated, and the large volume of published work, in combination with varying accessibility, makes collating data difficult, time-consuming, and resource intensive (Thessen, Cui & Mozzherin, 2012). For instance, a comprehensive review (Oficialdegui, Sánchez & Clavero, 2020) examined the global spread of Procambarus clarkii (Girard, 1852) identifying occurrences in over 40 countries. In the subsequent four years, the documented non-native populations have expanded their ranges and additional invasive populations have emerged in other countries previously deemed free of P. clarkii (Lipták et al., 2024).

A comprehensive, large-scale, and amendable database on crayfish distributions should be prioritized to enable quick reporting for accurate assessments of, and predictions about, the evolution of the distributions and the conservation status of these species (Reaser, Frey & Meyers, 2019). This database should facilitate tracking the dynamic ranges of both native and non-native species while being rigorously checked, maintained, and ultimately not subject to inaccuracies and biases of, e.g., citizen science data and non-scientific reports (Troudet et al., 2017; Chandler et al., 2017). Such a database could help scientists and stakeholders make more accurate assessments of the native or non-native status of populations by allowing the identification of possible migration patterns and the most plausible scenarios for the range expansions of some species. Moreover, the efforts of analyzing the spread of pathogens to anticipate the direction and speed of their range expansions and their effects on various species are still in the pioneering stage and require accurate mapping through collaborative efforts among researchers from various fields of expertise (Fišer, 2019; Strand et al., 2019; Marques et al., 2021). For this purpose, and to fill these knowledge gaps in the field of astacology, we introduce the World of Crayfish™ (WoC™). This e-portal provides an overview of the worldwide distribution of freshwater and semi-terrestrial crayfish and their most prominent pathogen, A. astaci (Martínez-Ríos, Martín-Torrijos & Diéguez-Uribeondo, 2023; Strand et al., 2023), laying the groundwork to address other problematic crustacean pathogens in the near future.

The WoC database and platform provide a valuable resource for addressing current data deficiencies by gathering and curating data from around the world, delivering distributional maps, and centralizing datasets. In addition, the platform provides an alternative repository for datasets from new publications or indeed prior to publication by offering a unique accession code for each available record. In doing so, WoC overcomes the publication lag time in reporting occurrence data. As WoC accumulates larger volumes of both temporal and spatial data, it will become increasingly relevant for agencies (e.g., the International Union for Conservation of Nature, CABI Compendium, European Alien Species Information Network) responsible for assessing and updating the conservation status of crayfish species and monitoring the changing distributions of native and non-native crayfish. Moreover, WoC is built to enable the implementation of additional features and improvements as data availability increases, opening new opportunities for state-of-the-art approaches to ecological modeling of crayfish and crayfish plague occurrences. This initiative is expected to boost and assist research and the publication of faunistic-focused studies on crayfish contributing to to their conservation and management.

Methods

Description

World of Crayfish is a platform that can be accessed at https://world.crayfish.ro/. The platform has an interface that welcomes the user with a brief description of its purpose, along with an introduction to the team behind the project and video instructions to help users understand and navigate the site. New data can be submitted through a downloadable EXCEL template accompanied by an instruction guide, enabling the platform to host continuously updated maps. Each data contributor can choose a restriction level prior to submitting new records to WoC. This feature permits that, if a high level of protection (e.g., threatened status of a particular crayfish species or distribution data prior publication) is selected, the exact location (and associated data) is hidden for non-registered users, but those records still contribute to the coarser-scale grid map visualization to ensure the best coverage of a species distribution. A technical review of new datasets is conducted by administrators using an offline preliminary check of the locations’ accuracy (and other additional provided information) in accordance with the original sources. The logical workflow is presented in Fig. 1.

Figure 1 Logical workflow.

Schematic representation of the logical workflow in and around the World of Crayfish environment, detailing potential data origins and how the data is processed by WoC before being provided to the end user. The figure was created using Inkscape (https://inkscape.org).

Table 1 The World of Crayfish database fields’ structure, also available as a template for contributors (NRU –non-registered users, and RU –registered users).

Section	Title	Brief description	Available for	
			NRU	RU	
Source information	DOI	Digital object identifier, a mandatory field (if available).	Yes	Yes	
URL	Uniform Resource Locator, the address of a given resource on the Web.	Yes	Yes	
Citation	Full reference for the source (for articles), APA format. For unpublished data, the contributors may ask for unique WoC database identifier.	Yes	Yes	
Geolocation	X	Latitude, in decimal degree, WGS 84 datum.	No	Yes	
Y	Longitude, in decimal degree, WGS 84 datum.	No	Yes	
Accuracy	Specifies how precise the provided coordinates are in relation to the place of field observation. High: data sourced from field records, maps at a hydrographic level, toponymy with high detail. Low: data sourced from continental or national grid systems, estimated locations based on toponymy with low detail.	No	Yes	
Crayfish	Crayfish scientific name	A mandatory field for the scientific name of the observed crayfish species. Exceptionally empty, when a record of Aphanomyces astaci without host (i.e., by eDNA detection) is stored.	Yes	Yes	
Status	Native: a crayfish species that naturally occurs in a specific region due to natural evolution over time. Non-native: a crayfish species (or associated pathogen) that has arrived facilitated by non-natural actions in a region where it does not belong due to natural evolution over time. Introduced: a crayfish species (or associated pathogen) that has arrived facilitated by human-mediated translocation. Type locality: if indexing the paper in which a species has been initially described. These categories in the table are not value-laden.	Yes	Yes	
Year of observation	The year when the observation of the crayfish species was made.	Yes	Yes	
NCBI COI accession code	The accession code in NCBI‘s GenBank for COI sequence if such data is available. Comma separated, if multiple.	No	Yes	
NCBI 16S accession code	The accession code in NCBI‘s GenBank for 16S sequence if such data is available. Comma separated, if multiple.	No	Yes	
NCBI SRA accession code	The Sequence Read Archive in NCBI‘s GenBank if such data is available. Comma separated, if multiple.	No	Yes	
Claim extinction	If confident that a crayfish population has completely disappeared this option is available. This selection will discard the spot from the time series maps starting with its year of record.	No	Yes	
Associated pathogens & symbionts	Pathogen/symbiont present	The scientific name of a crayfish associated pathogen (i.e., A. astaci) or symbiont (i.e., Branchiobdella sp.), if any, on a crayfish specimen (data stored in Crayfish section).	Yes	Yes	
Pathogen/symbiont detection method	Contributor declaration of the method used for detection of the pathogen/symbiont: Molecular/Microscopic/eDNA/any other.	Yes	Yes	
Pathogen/symbiont genotyping method	If available: Chitinase sequencing/Microsatellites/mtDNA sequencing/AFLP/RFLP/RAPD/Anonymous nuclear markers. Comma separated, if multiple.	Yes	Yes	
Genotype group	If available: A/B/D/E/Up/other described.	Yes	Yes	
Haplotype	If available: d1/d2/other described.	Yes	Yes	
Year of observation	The year when the samples were taken.	Yes	Yes	
Additional information	Comments	Any optional additional information that might be worth mentioning.	Yes	Yes	
Confidentiality level	The degree of public availability that the contributor has chosen for the data. No restrictions: data will be displayed with the exact location for all users. Regular restrictions: 20 km hexagonal shape displayed for NRU, exact location display for RU. High restrictions: data displayed in a 20 km hexagonal shape for all users.			
Contributor	The name of the contributor holder to the database for a respective record.	Yes	Yes	

Database

The database is structured in a single table containing the information, formatted in the same manner as the EXCEL template file for data contributors. The database fields are structured in five sections (Table 1). The first section is dedicated to the source of information, including the Digital Object Identifier (DOI), accompanied by the URL where the article/dataset can be found (if a DOI is not available) and the citation reference in APA format. The second section consists of location data (coordinates: X–latitude and Y–longitude, in decimal degrees, WGS84 datum), accompanied by accuracy (ensuring that the provided locations correspond to the place of a field observation). The third section centralizes the scientific species name (according to the taxonomy available in World Register of Marine Species, WoRMS, accessed on 14.03.2024, https://www.marinespecies.org/), its status at the locality (native vs. non-native or introduced), based on expert assessment in the original source of data records (sensu Soto et al., 2024), and a specific “type locality” mark, if indexing the paper in which a species has been originally described, year of observation, and a set of National Center for Biotechnology Information (NCBI) GenBank accession codes, if available (COI, 16S, SRA). Where there is a high level of confidence that a crayfish population has completely disappeared, this section can mark “extinct” which will discard the spot from the time series maps starting with the year of a scientifically documented population extinction record. The fourth section is dedicated to recording details of the crayfish associated pathogens or symbionts (if any) identified at the specified coordinates of the crayfish record, alongside pathogen detection and genotyping methods, and eventual genotype and haplotype classification. If detection of the pathogen or symbiont was confirmed by environmental DNA analysis, then sample coordinates are registered instead of a crayfish host. Finally, the fifth section provides any additional information that might be worth mentioning, including the contributor’s name and a confidentiality level that may establish restrictions for data. The confidentiality level will be displayed and downloaded with the associated location for users.

Each record is stored under a unique ID. The EXCEL template has a corresponding table in the PostgreSQL/PostGIS database allowing new contributions to be added easily. The corresponding database table has the same attributes as the EXCEL table but also a PostGIS specific geometry column so that the location data is easily managed by software. The data from the database is then read by the QGIS Server and served through the WoC platform.

Administrators and users

The user registration system (upstreamed in the Open-Source code) was developed so that a new user can apply, and receive access approval, based on a specific workflow and requirements setup in the administration side of the platform. New users are reviewed, approved and assigned custom security privileges by the platform administrators who get notified by e-mail of the new sign-up. The platform has an administrator interface that is used to assign granular security privileges for any users: view/edit privileges at project/theme level, layer level, column level or feature level. Feature level permissions are assigned using two kinds of filters: data source level - SQL “WHERE” filters or QGIS Server level - QGIS Expressions. Therefore, the distribution maps’ view is layered with two levels of access: (i) non-registered users (NRU), and (ii) registered users (RU). New users are accepted based on a declaration that can attest to professional interest.

Display

From the database, the platform displays the information about the selected taxon using OpenStreetMap (https://www.openstreetmap.org) as a basemap. For NRU, records are displayed as hexagons, with each edge measuring 20 km. Crayfish species occurrences are aggregated per cell to protect the information on the exact location of the crayfish for conservation purposes. We generated a global hexagonal cell grid using the QGIS platform “Create Grid” algorithm available in the QGIS Processing framework and saved it as a distinct table. To visually distinguish the different occurrences per species, the Jenks natural breaks classification method was used for grid display. For each species, we correlated the number of occurrences with the opacity value for each cell with a minimum of one occurrence. The opacity scale ranges from 20 (partially opaque cell) to 100 (fully opaque cell). In contrast, RU, once logged in, may also see a dotted map display with the exact crayfish locations shown on the map according to the coordinates stored in the database. One or multiple taxa can be inspected simultaneously and will be automatically allocated different colors, either with hexagons or dot shapes displayed. Aphanomyces astaci records can be displayed individually, or together with those of crayfish species.

Attribute data

From the WoC database, a set of associated data with the record(s) in the hexagon (for NRU) or dot (for RU), is displayed. Attribute data lists species names, status (Native/Non-native), number of observations in the selected hexagon (for NRU), year of observation, bibliographic references, and the names of the persons who indexed the data in WoC. The bibliographic reference, linked to a DOI or similar URL, will send the user directly to the source paper. Using specific geospatial standards, the attribution of information for each geospatial feature is accessed using a Web Map Service GetFeatureInfo request.

Export

The platform can generate maps and tables to be exported, available for the RU only, listing: GPS coordinates in WGS84 format (if the taxon is not specifically restricted, at the contributor’s request), accuracy of the localities, the countries in which the species is present, basic statistics on the number of records, and the related source literature. Also, the platform can be used as an alternative to dataset repositories for new publications as it provides unique, identifiable, searchable, and sharable ID (accession code) for each record. The user can access the data through standard geospatial services—OGC (Open Geospatial Consortium) services: WMS (Web Map Service) and WFS (Web Feature Service). Any RU has access to the data through the WFS endpoint according to their privileges, providing the possibilities of download, visualization and embedding into other applications (e.g., QGIS, ArcGIS, Geoserver). For instance, users can access the WMS at: https://map.crayfish.ro/ows/crayfish/?SERVICE=WMS&REQUEST=GetCapabilities.

Figure 2 Overview of the collected data in the World of Crayfish platform, and various centralisation of records of natives versus non-native (introduced status included).

(A–B) Crayfish species records at continental level (type locality and Aphanomyces astaci highlighted), (C–D) country level record counts, and (E–F) major hydrographic basins record counts. The lower panel integrates data for the best-represented species. The collage was created in Inkscape (https://inkscape.org) using maps generated in ArcGIS Pro (ESRI, Redlands) with freely available basemap layers: (A–B) world continents from ESRI (https://services.arcgis.com/P3ePLMYs2RVChkJx/arcgis/rest/services/World_Continents/FeatureServer), (C–D) administrative boundaries at country level from EuroGeographics (https://ec.europa.eu/eurostat/web/gisco/geodata/administrative-units/countries), and (E–F) major hydrological basins from the Food and Agriculture Organization of the United Nations (https://data.apps.fao.org/catalog//iso/7707086d-af3c-41cc-8aa5-323d8609b2d1). Crayfish silhouettes were drawn by Lucian Pârvulescu.

Data records

The key features offered by WoC are (1) the compilation of global records into a single accessible platform, enabling users to gain a comprehensive understanding of the distributions of many species of an important freshwater taxonomic group, and (2) the ability to grow by serving as a data repository for newly published studies on freshwater crayfish or the crayfish plague pathogen. Here we present an informative overview of the database at the time of the publication (Fig. 2, and Tables S1–S3), with 105,611 records indexed from 427 taxa (species, subspecies or species complex; see Table S1). Of these records, 72.9% (n = 77,062) are of high geolocation accuracy. In the current dataset, the North American continent has the highest diversity (including non-native species) with 390 taxa indexed and 51,940 stored records (49.2%). For a detailed count per country, we refer readers to Table S2. The Mississippi - Missouri River Basin holds almost one quarter of the global occurrence records (23,301; 22.1%; Table S3). Almost half (48.1%) of the total observations currently held by WoC (n = 50,838) are ascribed to non-native species, with the most widely reported non-native species being P. clarkii with 34,587 records (32.8%) across six continents. The second most reported species outside its native range is Faxonius limosus (Rafinesque, 1817) with 6,136 records (5.8%). Currently, there are 428 data records associated with A. astaci, which are mostly spread over continental Europe.

With the severe decline of native crayfish populations and the rapid spread of non-native crayfish (Richman et al., 2015), WoC is expected to be the go-to access point for the rapid and reliable interpretation of spatial and temporal dynamics of crayfish species globally. The easy access to contributing references is helpful not only for new studies, but also for making the data from existing scientific publications available to a broader audience. Moreover, the contribution of international collaborators allows WoC to reach crayfish gray literature, often written in local languages and concerning local context (Hannah et al., 2024). The accessibility and utilization of gray literature, although often neglected, holds paramount importance in nature conservation (Amano et al., 2021; Amano et al., 2023). The platform offers the possibility to revise one or multiple indexed locations to list a population as extinct, with 126 such situations already indexed. This function allows for time series interruptions if new data suggest that populations from a previously published resource have disappeared, indicating a potential focal point for various ecological pressures (for example, habitat degradation or destruction). In contrast, once a regular location point is entered, it remains “forever” visible in the time series from the date of the original observation stored in the database.

By their very nature, global-scale analyses require large datasets, but distributional data are of limited value if not associated with geospatial ecological variables specifically designed to describe the hydrographic networks (Domisch, Amatulli & Jetz, 2015). State-of-the-art modeling approaches offer a means to explore and uncover previously unrecognized ecological relationships, thereby reducing uncertainty and validating the significance of outputs (Shen et al., 2023). The WoC platform has been designed to address these challenges by storing locations based on the accuracy of each data point, ensuring high-quality representation of the water bodies where the crayfish reside. This approach enables us to filter data appropriately for different modeling purposes. The platform could provide opportunities to assess, for instance, habitat selection patterns and post-glacial migration routes taken by crayfish species and identify the glacial refugia these crayfish may have used during the last Ice Age; which can be useful for understanding modern day distributions and predicting future changes in distributions in relation to climate change and other environmental variables (Guiasu, Barr & Dunham, 1996; Guiaşu & Labib, 2021). We advocate for the use of geospatial data designed to describe dendritic river networks instead of landscape-based approaches, as they provide a more ecologically relevant representation of crayfish habitats (Fetzner & Crandall, 2003). The next level in the development of the WoC platform is to add a feature whereby registered users will be able to download customizable datasets to support their modeling procedures.

Technical validation

The ability to visualize geographic ranges of multiple species and their environmental conditions is a critical resource for ecological and conservation research aiming to assess an organism’s response to change (Villero et al., 2017). Centralized resources will improve capacity for spatially explicit management and policy decision-making regarding endemic, threatened, and non-native crayfish (Taylor et al., 2019). It is important to ensure that such data are based on rigorous investigations to avoid potential errors and inaccuracies. The distinctions and boundaries between native and non-native ranges, particularly in areas where these ranges are adjacent to each other and not separated by any obvious environmental barriers, can often be difficult to establish and decisions on the native or non-native status of species can be subjective (Pereyra & Guiaşu, 2020). Comprehensive, regularly updated, and professionally curated datasets on the distributions of species hold the promise of more accurate diagnoses regarding the status of these species and the evolution of their ranges over time. Raw data, such as those obtained from citizen science online databases, are generally useful, but need expert validation as they may be prone to errors resulting from taxonomic confusion, species misidentification, and mistakes in recording locations, among others (Clare et al., 2019; Lipták et al., 2024). In the particular case of crayfish, expert-validated data are more likely to be accurate and reliable since they are typically subjected to rigorous taxonomic scrutiny, which is essential for accurate species identification (Jiménez-Valverde, Lobo & Hortal, 2008). In some cases, additional molecular analyses are used to confirm taxonomic identifications, further increasing the accuracy of the data (Bystriakova et al., 2012). Nevertheless, there are also some disadvantages associated with data from published sources as, for example, the lag time between data being obtained and published may be slow, limiting its usefulness for time-sensitive research. Data from published sources can also be difficult to centralize due to the diverse publication access policies of different journals and the heterogeneity of databasing formats (Reichman, Jones & Schildhauer, 2011).

WoC’s maps provide an advantage over drawing from the literature by eliminating the time-consuming, and often repeated, effort for every new study. Our indexed data are delivered in a unitary format, with emphasis on geographic locations in WGS84 datum and validated accuracy, which greatly eases and streamlines the processes required for modeling input data. The known level of accuracy allows data filtering, saving time, and enabling quality datasets. It is important to note that, when visualizing large-scale distribution maps, accuracy is not always prioritized; hence, our map can display all levels of data accuracy. We provide a confidence rating through the accuracy attribute in our downloadable datasets, allowing modelers to make informed decisions on the preciseness of the data they use. When developing models, the presence of redundancy and imbalances in sampling can impact the overall quality of the model. Therefore, we recommend that researchers consider spatial re-sampling to ensure uniform geographical coverage in their selected areas (Guisan & Zimmermann, 2000). Species-associated data issues can also be related not only to taxonomic errors (Meier & Dikow, 2004) but also to inaccurate or incomplete spatial and temporal information (Meyer, Weigelt & Kreft, 2016). While misidentification should be a lesser issue for data from peer-reviewed articles, spatial errors can occur and need clarification (Graham et al., 2008; Führding-Potschkat, Kreft & Ickert-Bond, 2022). One of the most frequent issues observed while indexing papers for WoC was the insufficient standardization of spatial data formatting and overall dataset heterogeneity. When coordinates of locations were supplied in publications (usually as supplementary material) and not just maps, these were sometimes lacking important information like the type of geographic coordinates used or zones for Universal Transverse Mercator (UTM) grid derived systems. These situations generate confusion for most users without GIS training. A lack of more detailed and precise information on the coordinate reference systems may lead to significant positional errors with a shift from the original position between zero and more than 5,000 m (Wieczorek, Guo & Hijmans, 2004; Guralnick et al., 2006). Even though a high correlation between models derived from the Global Biodiversity Information Facility (GBIF) and those from expert data is often found (Führding-Potschkat, Kreft & Ickert-Bond, 2022), expert data did not require processing. While possibly true for some taxa, assembling several expert datasets for crayfish still requires some degree of homogenization and a comprehensive review for completeness.

Finally, there is a concerning trend of declining interest in faunistic studies at the level of high-impact scientific journals (Ejsmont-Karabin, 2019). Faunistic studies may be of more interest to some governmental organizations, which only target certain species, such as those listed in legislation either because they may be vulnerable or endangered or because they may be regarded as invasive. This potential bias leaves a large gap in scientific knowledge and the conservation of biodiversity at a global level, since most other species are largely ignored (Valdecasas & Camacho, 2003; Rodríguez et al., 2015). Data curation with the aid of automated information retrieval from published literature is a promising avenue for near real-time databasing (Kopperud, Lidgard & Liow, 2022). Screening tools and automatic data collection from the scientific literature based on natural language processing to extract the locations of crayfish species with text-mined occurrences, and solving language barriers by machine linguistic translation, are essential for sourcing large scale biodiversity data (Amano et al., 2023; Hannah et al., 2024). In addition, features for automated population genetics and phylogenetic analyses are plausible, based on freely accessible data from public genetic databases and popular open-source packages implemented using the R programming language (RCoreTeam, 2024). All these features may represent future opportunities to improve the WoC platform and reinvigorate the interest for faunistic studies not only on crayfish, but also on a variety of other taxa.

Usage notes

World of Crayfish itself is a repository database. In addition to the repository files, these data are also available via the interactive web platform (https://world.crayfish.ro/). Site access is free, however exact location of crayfish species is restricted to prevent commercial exploitation of sensitive species. The ‘fairness’ of WoC has been assessed using FairShake, which provided a comprehensive evaluation confirming that the datasets adhere to the FAIR principles (Wilkinson et al., 2016; Barker et al., 2022) through rigorous metrics and criteria, ensuring high standards of data management and usability. The WoC is a word and figurative trademark, pending registration (No. M2024/08475 of 16.09.2024) at the Romanian National Office for Intellectual Property, at the time of this paper’s publication. The request for trademark award is done under Class 42 of the Nice Classification (World Intellectual Property Organization, 2022). The award of the national trademark certification will be followed with a request for international protection at the World Intellectual Property Office in accordance with the Madrid System. However, we encourage citation of those sources associated with any record in WoC, and the WoC platform should be cited as the tool for delivering multi-source compilations and various outputs.

Code availability

WoC was developed using Open-Source software g3w-suite (https://github.com/g3w-suite/), a Django based QGIS platform and PostgreSQL/Postgis. The code written for WoC (e.g., https://github.com/g3w-suite/g3w-admin/pull/610) has been upstreamed to allow broader access and to reduce the maintenance burden.

Conclusions

The primary product offered by WoC is knowledge, facilitated through machine-driven querying of scientific data, aligning with the Digital: Green: Nature trend. Looking ahead, this project has the potential to elevate the conceptual use and utility of biodiversity databases. WoC envisions a platform populated exclusively by experts and their study data, developed with the highest standards and using the most advanced technologies, fostering interdisciplinarity between biological sciences and Geographic Information Systems.

To date, 98.5% of the data hosted on WoC is sourced from scientific papers, data reports, and museum databases. Additionally, WoC offers a repository function for further publication with 1,444 (1.5%) records from otherwise unpublished datasets. The database also provides, where available, the specific GenBank accession codes for the most common DNA sequences from genetic studies on crayfish, with 2,065 for CO1, 1,375 for 16S, and 15 SRA records already being available in WoC. The platform also features a function to filter species records based on the status attribute, listed as “type locality”. This functionality, which presently returns 16 type localities, allows users to efficiently access hard-to-find information, such as original formal species descriptions from the scientific literature, and visualize this information on a map. To protect sensitive species information, the exact geographic location of a species is only available to RUs, whereas NRUs can only view the records within hexagonal areas.

Crayfish exhibit a distinctive ecology, inhabiting a wide range of aquatic and semi-terrestrial habitats (Neculae et al., 2024). The distributional data hosted by WoC, combined with geospatial data specifically adapted to this diversity of habitats (e.g., Pârvulescu et al., 2016; Şandric et al., 2019), will improve analyses of the distributional patterns of crayfish. The future integration of statistical and data processing methods will significantly enhance our platform’s capabilities and further biogeographical research. Additionally, we plan to improve user accessibility and interaction through advanced IT programming, culminating in an Artificial Intelligence interaction across various levels (data collection and curation, results interpretation, reporting), making it easier for users to navigate, analyse, and utilize the information available on the platform. With certain future potential, WoC is the first global platform created for scientists, society, and stakeholders, specifically to support the conservation and management of crayfish as an important and historically under-resourced taxon.

Supplemental Information

Supplemental Information 1 Records count per crayfish taxa integrated in WoC platform

Supplemental Information 2 Crayfish and A. astaci records integrated in WoC platform count per country

Supplemental Information 3 Crayfish and A. astaci records per major hydrographic basins

Crayfish and A. astaci records integrated in WoC platform count per major hydrographic basins.

The authors wish to extend their gratitude, in advance, to the forthcoming contributors who will generously contribute to the growth of WoC. The invaluable support from these data providers is essential, as their contributions are instrumental for enhancing the richness and value of this platform. We appreciate the comments provided from three anonymous reviewers which served to improve the manuscript.

Additional Information and Declarations

Competing Interests

Author Contributions

Patent Disclosures

Data Availability

The authors declare there are no competing interests.

Mihaela C. Ion conceived and designed the experiments, performed the experiments, analyzed the data, prepared figures and/or tables, authored or reviewed drafts of the article, data curation, and approved the final draft.

Caitlin C. Bloomer conceived and designed the experiments, performed the experiments, authored or reviewed drafts of the article, data curation, and approved the final draft.

Tudor I. Bărăscu conceived and designed the experiments, performed the experiments, authored or reviewed drafts of the article, iT support and development, and approved the final draft.

Francisco J. Oficialdegui conceived and designed the experiments, authored or reviewed drafts of the article, database, and approved the final draft.

Nathaniel F. Shoobs conceived and designed the experiments, authored or reviewed drafts of the article, database, and approved the final draft.

Bronwyn W. Williams conceived and designed the experiments, authored or reviewed drafts of the article, database, and approved the final draft.

Kevin Scheers conceived and designed the experiments, authored or reviewed drafts of the article, database, and approved the final draft.

Miguel Clavero conceived and designed the experiments, authored or reviewed drafts of the article, database, and approved the final draft.

Frédéric Grandjean conceived and designed the experiments, authored or reviewed drafts of the article, database, and approved the final draft.

Marc Collas conceived and designed the experiments, authored or reviewed drafts of the article, database, and approved the final draft.

Thomas Baudry conceived and designed the experiments, authored or reviewed drafts of the article, database, and approved the final draft.

Zachary Loughman conceived and designed the experiments, authored or reviewed drafts of the article, database, and approved the final draft.

Jeremy J. Wright conceived and designed the experiments, authored or reviewed drafts of the article, database, and approved the final draft.

Timo J. Ruokonen conceived and designed the experiments, authored or reviewed drafts of the article, database, and approved the final draft.

Christoph Chucholl conceived and designed the experiments, authored or reviewed drafts of the article, database, and approved the final draft.

Simone Guareschi conceived and designed the experiments, authored or reviewed drafts of the article, database, and approved the final draft.

Bram Koese conceived and designed the experiments, authored or reviewed drafts of the article, database, and approved the final draft.

Zsombor M. Banyai conceived and designed the experiments, authored or reviewed drafts of the article, database, and approved the final draft.

James Hodson conceived and designed the experiments, authored or reviewed drafts of the article, database, and approved the final draft.

Margo Hurt conceived and designed the experiments, authored or reviewed drafts of the article, database, and approved the final draft.

Katrin Kaldre conceived and designed the experiments, authored or reviewed drafts of the article, database, and approved the final draft.

Boris Lipták conceived and designed the experiments, authored or reviewed drafts of the article, database, and approved the final draft.

James W. Fetzner conceived and designed the experiments, authored or reviewed drafts of the article, database, and approved the final draft.

Tommaso Cancellario conceived and designed the experiments, authored or reviewed drafts of the article, database, and approved the final draft.

András Weiperth conceived and designed the experiments, authored or reviewed drafts of the article, database, and approved the final draft.

Jạnis Birzaks conceived and designed the experiments, authored or reviewed drafts of the article, database, and approved the final draft.

Teodora Trichkova conceived and designed the experiments, authored or reviewed drafts of the article, database, and approved the final draft.

Milcho Todorov conceived and designed the experiments, authored or reviewed drafts of the article, database, and approved the final draft.

Maksims Balalaikins conceived and designed the experiments, authored or reviewed drafts of the article, database, and approved the final draft.

Bogna Griffin conceived and designed the experiments, authored or reviewed drafts of the article, database, and approved the final draft.

Olga N. Petko conceived and designed the experiments, authored or reviewed drafts of the article, database, and approved the final draft.

Ada Acevedo-Alonso conceived and designed the experiments, authored or reviewed drafts of the article, database, and approved the final draft.

Guillermo D’Elía conceived and designed the experiments, authored or reviewed drafts of the article, database, and approved the final draft.

Karolina Śliwińska conceived and designed the experiments, authored or reviewed drafts of the article, database, and approved the final draft.

Anatoly Alekhnovich conceived and designed the experiments, authored or reviewed drafts of the article, database, and approved the final draft.

Henry Choong conceived and designed the experiments, authored or reviewed drafts of the article, database, and approved the final draft.

Josie South conceived and designed the experiments, authored or reviewed drafts of the article, database, and approved the final draft.

Nick Whiterod conceived and designed the experiments, authored or reviewed drafts of the article, database, and approved the final draft.

Katarina Zorić conceived and designed the experiments, authored or reviewed drafts of the article, database, and approved the final draft.

Peter Haase conceived and designed the experiments, authored or reviewed drafts of the article, database, and approved the final draft.

Ismael Soto conceived and designed the experiments, authored or reviewed drafts of the article, database, and approved the final draft.

Daniel J. Brady conceived and designed the experiments, authored or reviewed drafts of the article, database, and approved the final draft.

Phillip J. Haubrock conceived and designed the experiments, authored or reviewed drafts of the article, database, and approved the final draft.

Pedro J. Torres conceived and designed the experiments, authored or reviewed drafts of the article, database, and approved the final draft.

Denis Şadrin conceived and designed the experiments, authored or reviewed drafts of the article, database, and approved the final draft.

Pavel Vlach conceived and designed the experiments, authored or reviewed drafts of the article, database, and approved the final draft.

Cüneyt Kaya conceived and designed the experiments, authored or reviewed drafts of the article, database, and approved the final draft.

Sang Woo Jung conceived and designed the experiments, authored or reviewed drafts of the article, database, and approved the final draft.

Jin-Young Kim conceived and designed the experiments, authored or reviewed drafts of the article, database, and approved the final draft.

Xavier H.C. Vermeersch conceived and designed the experiments, authored or reviewed drafts of the article, database, and approved the final draft.

Maciej Bonk conceived and designed the experiments, authored or reviewed drafts of the article, database, and approved the final draft.

Radu Guiaşu conceived and designed the experiments, authored or reviewed drafts of the article, database, and approved the final draft.

Muzaffer M. Harlioğlu conceived and designed the experiments, authored or reviewed drafts of the article, database, and approved the final draft.

Jane Devlin conceived and designed the experiments, authored or reviewed drafts of the article, database, and approved the final draft.

Irmak Kurtul conceived and designed the experiments, authored or reviewed drafts of the article, database, and approved the final draft.

Dagmara Błońska conceived and designed the experiments, authored or reviewed drafts of the article, database, and approved the final draft.

Pieter Boets conceived and designed the experiments, authored or reviewed drafts of the article, database, and approved the final draft.

Hossein Masigol conceived and designed the experiments, authored or reviewed drafts of the article, database, and approved the final draft.

Paul R. Cabe conceived and designed the experiments, authored or reviewed drafts of the article, database, and approved the final draft.

Japo Jussila conceived and designed the experiments, authored or reviewed drafts of the article, database, and approved the final draft.

Trude Vrålstad conceived and designed the experiments, authored or reviewed drafts of the article, database, and approved the final draft.

David V. Beresford conceived and designed the experiments, authored or reviewed drafts of the article, database, and approved the final draft.

Scott M. Reid conceived and designed the experiments, authored or reviewed drafts of the article, database, and approved the final draft.

Jiří Patoka conceived and designed the experiments, authored or reviewed drafts of the article, database, and approved the final draft.

David A. Strand conceived and designed the experiments, authored or reviewed drafts of the article, database, and approved the final draft.

Ali S. Tarkan conceived and designed the experiments, authored or reviewed drafts of the article, database, and approved the final draft.

Frédérique Steen conceived and designed the experiments, authored or reviewed drafts of the article, database, and approved the final draft.

Thomas Abeel conceived and designed the experiments, authored or reviewed drafts of the article, database, and approved the final draft.

Matthew Harwood conceived and designed the experiments, authored or reviewed drafts of the article, database, and approved the final draft.

Samuel Auer conceived and designed the experiments, authored or reviewed drafts of the article, database, and approved the final draft.

Sandor Kelly conceived and designed the experiments, authored or reviewed drafts of the article, database, and approved the final draft.

Ioannis A. Giantsis conceived and designed the experiments, authored or reviewed drafts of the article, database, and approved the final draft.

RafałMaciaszek conceived and designed the experiments, authored or reviewed drafts of the article, database, and approved the final draft.

Maria V. Alvanou conceived and designed the experiments, authored or reviewed drafts of the article, database, and approved the final draft.

Önder Aksu conceived and designed the experiments, authored or reviewed drafts of the article, database, and approved the final draft.

David M. Hayes conceived and designed the experiments, authored or reviewed drafts of the article, database, and approved the final draft.

Tadashi Kawai conceived and designed the experiments, authored or reviewed drafts of the article, database, and approved the final draft.

Elena Tricarico conceived and designed the experiments, authored or reviewed drafts of the article, database, and approved the final draft.

Adroit Chakandinakira conceived and designed the experiments, authored or reviewed drafts of the article, database, and approved the final draft.

Zanethia C. Barnett conceived and designed the experiments, authored or reviewed drafts of the article, database, and approved the final draft.

Ştefan G. Kudor conceived and designed the experiments, performed the experiments, authored or reviewed drafts of the article, data curation, and approved the final draft.

Andreea E. Beda conceived and designed the experiments, authored or reviewed drafts of the article, iT support and development, and approved the final draft.

Lucian Vîlcea conceived and designed the experiments, performed the experiments, authored or reviewed drafts of the article, iT support and development, and approved the final draft.

Alexandru E. Mizeranschi conceived and designed the experiments, authored or reviewed drafts of the article, literature review, and approved the final draft.

Marian Neagul conceived and designed the experiments, authored or reviewed drafts of the article, iT support and development, and approved the final draft.

Anton Licz conceived and designed the experiments, authored or reviewed drafts of the article, iT support and development, and approved the final draft.

Andra D. Cotoarbă conceived and designed the experiments, authored or reviewed drafts of the article, and approved the final draft.

Adam Petrusek conceived and designed the experiments, authored or reviewed drafts of the article, literature review, and approved the final draft.

Antonín Kouba conceived and designed the experiments, authored or reviewed drafts of the article, literature review, and approved the final draft.

Christopher A. Taylor conceived and designed the experiments, authored or reviewed drafts of the article, project supervision, coordinate data collection, facilitate collaborations, and approved the final draft.

Lucian Pârvulescu conceived and designed the experiments, performed the experiments, analyzed the data, prepared figures and/or tables, authored or reviewed drafts of the article, project management, coordinate data collection, facilitate collaborations, and approved the final draft.

The following patent dependencies were disclosed by the authors:

The World of Crayfish (WoC) trademark is pending registration at the Romanian National Office for Intellectual Property. The request for trademark award is done under Class 42 of the Nice Classification (World Intellectual Property Organization, 2022). The award of the national trademark certification will be followed with a request for international protection at the World Intellectual Property Office in accordance with the Madrid System.

The following information was supplied regarding data availability:

The data is available in the Supplemental Files.

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
