# Peer review of "World of Crayfish™: a web platform towards real-time global mapping of freshwater crayfish and their pathogens"

_PeerJ, doi:10.7717/peerj.18229_

## Round 0.1 · original submission · Minor Revisions

Dear authors,

We kindly request that you carefully review the comments provided by the reviewers. Their valuable suggestions offer insights to enhance your manuscript. Incorporate their suggestions and carefully address all comments in your manuscript; it will significantly strengthen its content.

Reviewer 1 ·

Basic reporting

see attached

Experimental design

see attached

Validity of the findings

see attached

Annotated reviews are not available for download in order to protect the identity of reviewers who chose to remain anonymous.

·

Basic reporting

The World of Crayfish: A web platform towards real-time global mapping of freshwater crayfish and their pathogens article is write in a clear and unambiguous, professional English. Literature references are sufficient and description of background context is provided in clear scientific format. For the structure of article I recommend to make a distinction between Methods chapter is from line 268 to 360 and Results and discussions from 361 to 507 and it is very useful to have Conclusions to synthetize results and current and future benefits of your research.

Experimental design

The current article is presenting the needs of multiuser platform development for globally, regional and local, time series assessment of target species distribution. The proposed databases with spatial and temporal species distribution are organized in a web-GIS principles and can integrate data from citizen to expert integration. The research is in a Scope of the journal. The research question is well defined. The research performed is relevant & meaningful and will cover the knowledge gap related to interdisciplinarity from biologist with expertise in species identification and geoscientist with main expertise in mapping mechanism and process. To maintain the high technical standard I recommend to reformulate line 291, 436 and in Table 1 to reflect the parameters (latitude, longitude coordinate in decimal degree units, with datum WGS84) of geographic location expressed in geographic coordinate system no projection. Projection is a coordinate reference system (CRS) where the surface of the earth is projected on plan to have a common measured units like meters, kilometers and most used projection are UTM and ETRS89. Also I recommend use of OpenStreetMap like basemap not "information...projected on ..." in line 327 and 328. Methods are described with sufficient details to be replicated.

Validity of the findings

Data records and Technical validation chapters preset enough information to present the benefits of aggregated data findings and novelty of integrated analysis. Integrate of all relevant data for crayfish species distribution most of data already validated and published with local relevance are used for regional and global assessments on collaborative work platform.
The Conclusions chapter is needed.

Additional comments

The article is covering a relevant research and tools development for data sharing and collaborative work platform for scientific group with common interes.

Reviewer 3 ·

Basic reporting

Generally quite solid. I have some specific comments in my review.

Experimental design

Well.... difficult to comment on experimental design when there's no experiment. Maybe you guys should rename this section?

Again, satisfactory in this section.

Validity of the findings

A bit of an oddball of a MS since it's describing what is essentially a website. Saying that isn't to disparage the manuscript and especially not the website. I'm just saying that it doesn't really fit into traditional review parameters

Additional comments

I have a much more thorough review that will be attached as a separate file.

Annotated reviews are not available for download in order to protect the identity of reviewers who chose to remain anonymous.

---

## Round 0.2 · accepted · Accept

World of Crayfish™: A web platform towards real-time global mapping of freshwater crayfish and their pathogens article about the open source tool, marks a significant advancement in modern conservation planning and the management of freshwater biodiversity.

Reviewer 1 ·

Basic reporting

The basic reporting is clear with appropriate set up and referencing.

Experimental design

The aims and scope of the study are well within the framework of PeerJ and the paper makes a great addition to the literature.

Validity of the findings

This will be a solid resource that will hopefully gain traction over time and become a great resource to both crayfish folks but also conservation and freshwater researchers.

Additional comments

The authors have done a very nice job updating their manuscript and addressing my prior concerns.

·

Basic reporting

The authors are presenting the article in a clear an unambiguous, professional English. Literature references of current article are sufficient on contextual basis. The article respect the specific structure and also contain representative tables and figures. The authors are following the current trend on relevant data sharing publishing also in increase the accessibility and availability in a digital form.

Experimental design

The experimental design of WoC platform is in the form of current trend in field of biodiversity platforms to increases responsibility and scientific and managerial care for species with vulnerable status. The spatial distribution at global, regional and local scales are most important to have a clear view on species state trends. Methods used for development of Webgis platform publishing like web feature services (WFS) cover needs for integrate assessment of species distribution relevant for species trend dynamics in space and time.

Validity of the findings

no comment

Additional comments

I want to congratulate the authors for their efforts to develop the WoC and I really appreciate the collaborative work for doing this.